# Barriers and Facilitators of Online Grocery Services: Perceptions from Rural and Urban Grocery Store Managers

**DOI:** 10.3390/nu14183794

**Published:** 2022-09-14

**Authors:** Rachel Gillespie, Emily DeWitt, Angela C. B. Trude, Lindsey Haynes-Maslow, Travis Hudson, Elizabeth Anderson-Steeves, Makenzie Barr, Alison Gustafson

**Affiliations:** 1Family and Consumer Sciences Extension, College of Agriculture, Food and Environment, University of Kentucky, Lexington, KY 40506, USA; 2Department of Nutrition and Food Studies, Steinhardt School of Culture, Education, and Human Development, New York University, New York, NY 10003, USA; 3Department of Health Policy & Management, Gillings School of Global Public Health, University of North Carolina at Chapel Hill, Chapel Hill, NC 27599, USA; 4Department of Nutrition, College of Education, Health, and Human Sciences, University of Tennessee, Knoxville, TN 37996, USA; 5Gretchen Swanson Center for Nutrition, Omaha, NE 68154, USA; 6Department of Dietetics and Human Nutrition, College of Agriculture, Food and Environment, University of Kentucky, Lexington, KY 40506, USA; 7College of Nursing, University of Kentucky, Lexington, KY 40506, USA

**Keywords:** online grocery, grocery shopping, behavior, food environment, SNAP

## Abstract

Online grocery shopping has expanded rapidly in the U.S., yet little is known about the retailer’s perceptions of online grocery services, which can aid in the expansion of services. Furthermore, many barriers to online grocery utilization persist across geographic areas, especially among Supplemental Nutrition Assistance Program (SNAP)-authorized retailers. This study captured perceived barriers and facilitators of online grocery shopping for managers of SNAP-authorized retailers. Qualitative semi-structured interviews were conducted with managers (n = 23) of grocery stores/supermarkets in urban and rural areas across four different states: TN, KY, NC, and NY. Grocery store managers offering online ordering (n = 15) and managers from brick-and-mortar stores without online services (n = 8) participated in the interviews. Three primary themes emerged among managers offering online ordering: (1) order fulfillment challenges, (2) perceived customer barriers, and (3) perceived customer benefits. Among managers at brick-and-mortar locations without online services, four major themes emerged: (1) thoughts on implementing online shopping, (2) COVID-19 pandemic impacts, (3) competition with other stores, and (4) benefits of maintaining brick-and-mortar shopping. This study provides a deeper understanding of retailers’ experience and perceptions of online grocery services among stores authorized to accept SNAP benefits. This perspective is necessary to inform policies and enhance the evolving virtual food marketplace for SNAP customers.

## 1. Introduction

Online grocery shopping has expanded at an accelerated rate in many higher-income and affluent communities and also for households of low-income backgrounds [1]. In response to the COVID-19 pandemic, the United States Department of Agriculture (USDA) authorized the expansion of the Supplemental Nutrition Assistance Program (SNAP) Online Purchasing Pilot (OPP) [2]. The OPP was first authorized in the 2014 Farm Bill (PL 113-79) and “mandated a pilot be conducted to test the feasibility and implications of allowing retail food stores to accept SNAP benefits through online transactions.” SNAP recipients are allowed to purchase SNAP-eligible foods online from USDA-authorized vendors. Vendors are required to have online software that fits the USDA’s technology security requirements. Additionally, SNAP benefits cannot be used to pay for fees (e.g., delivery, service, membership) of any type.

Recently, the USDA expanded the online delivery options to include more smaller and mid-size stores to enhance the reach of SNAP customers with Electronic Benefit Transfer (EBT) cards in efforts to promote food access equity. Although there has been an expansion of online services, limited uptake of online grocery services by consumers due to a variety of barriers such as digital literacy, limited delivery services, and delivery fees [3,4,5] remains. Despite the individual-level factors that may prohibit the adoption of utilizing online grocery shopping services, several store-level features may also limit online shopping utilization by customers and stores. Reasons for limited utilization of this online feature are still being understood, adapted, and refined.

The potential benefits of online grocery shopping include improving access to food venues not typically utilized by customers due to physical distances, thus facilitating the extension of an individuals’ “neighborhood”, or food environment, across a larger area [6]. This may offer a wider variety of lower-cost food items to low-income consumers [7] living in areas with low availability of supermarkets in both urban and rural areas. As of 2021, the USDA Food Desert Locator identified nearly 13.5 million individuals, encompassing 10% of the total number of census tracts within the U.S., as having low access to healthful foods [8]. Online platforms offering delivery services aid in addressing the transportation barrier some individuals may face by minimizing travel to grocery stores as well as the time required for shopping [7,9]. A recent study indicated the strong potential for both rural households and SNAP households utilizing online grocery shopping to increase the overall quality of food among households [7]. This is important because these demographic populations have been designated by the National Institute of Health (NIH) as “vulnerable” due to the current profound disparities in health status in the U.S. [10]. However, there remains a need to better understand the operational capacity of this grocery shopping mechanism, as it has the potential to improve food access and dietary patterns among populations most in need of healthier and affordable food options.

As online grocery shopping has become more prominent, grocery store managers have modified departments, operations, and employee roles to ensure adequate supply, staffing, and technology are available to serve customers [11]. A conceptual framework, originally outlined by Khandpur et al., displayed in Figure 1, was recently developed to depict the growing realm of exposure and influence of the online grocery shopping landscape among consumers and retailers and the complex interactions that exist, thereby influencing consumer behaviors [12,13]. Domains of the framework include retailer-level and consumer-level attributes influencing behaviors progressing through the path-to-purchase phases of the online shopping process, which includes pre-shop, online shopping, pick-up or delivery, and post-shop. This consumer process, and the corresponding retailer-level activities, ultimately encompasses the quality of the experience in which the consumer engages in.

Recent work has focused on the consumer experience with online grocery services [7,14,15,16,17]. However, little is known about the barriers and facilitators of online grocery services from the retailers’ perspective. Other countries have explored fulfillment and distribution logistics, including challenges retailers face [18], yet little is known about the overall retailer perspective with online grocery services from pre- to post-shopping experience. Further, Martín, J. C., et al. (2019) have previously identified several gaps in online grocery research [19], yet the retailer perspective is not one that has been highlighted. Therefore, it is necessary to continue exploring each level that comprises the entirety of the retailer’s role, as grocery store managers play an important role in shaping the retail food environment and the customer experience when shopping for food, fulfilling online orders, managing inventory, and other key supply chain processes, both in-person and online, as outlined within the framework. Although the proposed consumer behavior conceptual framework encompasses both consumer and retailer-level behaviors, this study focuses primarily on the retailer level to fill an important gap in the literature: the perspective of suppliers. Through this innovative approach, we aim to build upon the concept of the different factors influencing consumer grocery shopping behaviors by providing a micro-lens into the broader macro-scale of the online food retail environment. This study of the retailer experience aims to shed light on the interactive nature of the manager experience and customer shopping experience, which have yet to be revealed [20]. Collectively, this will expand on the understanding of how suppliers navigate the online retail modality to fulfill consumer demand.

The current study addresses these gaps by exploring the experiences with online retail sales and operations from grocery managers of both brick-and-mortar and online retailers. The aims of this study are (1) to explore the barriers and facilitators of online shopping from the retailer perspective offering online grocery services and authorized to accept SNAP benefits online and (2) to explore barriers and feasibility of incorporating online services into brick-and-mortar grocery stores in both urban and rural communities currently without online shopping capabilities but authorized to accept SNAP benefits. These findings aim to help improve online shopping platforms and provide key data for policy improvement related to SNAP online authorizations in both rural and urban settings.

## 2. Materials and Methods 

In the following section, methods are explained for the approach taken for the current study. Section 2.1 explains the data collection study sites that were contacted to participate, including their geographical locations. Section 2.2 explains the interview process conducted in the semi-structured interviews. Finally, Section 2.3 discusses the analyses of the qualitative data including recording and transcribing of data, codebook development, and the processes of multi-coder processing.

### 2.1. Study Sites

Grocery store managers were contacted in the Fall of 2021 to participate in semi-structured interviews. Stores were purposefully included to comprise the store locations that participated in an 8-week online grocery intervention [21] or through existing community partnerships with grocery stores. To capture unique perspectives from various store sizes and online capacity, a targeted recruitment strategy was conducted based on the following criteria: store size, online ordering capacity, SNAP-authorized retailer, rural vs. urban setting, and type of store. Managers of 11 urban stores and 12 rural stores completed semi-structured in-depth interviews: 6 stores in Tennessee (TN) (2 urban and 4 rural), 5 in Kentucky (KY) (3 urban and 2 rural), 6 in North Carolina (NC) (2 urban and 4 rural), and 6 urban stores in New York (NY). Counties were classified as rural or urban based on the USDA’s rural-urban continuum (RUC) codes (3–8) [22]. Counties were selected based on availability of SNAP online ordering—pick-up or delivery options. Rural counties were selected with a designated RUC code of 6 or higher, while selected urban counties had a RUC code of 3–5. The study team also aimed to select grocery stores that were similar in number of cash registers and store square footage to maintain consistency in store operational capacity.

### 2.2. Interview Process

Grocery store managers were asked to participate in a 30–60-minute interview about experiences with online ordering, including barriers, facilitators, and preferences, in October and November 2021. A total of 53 managers were contacted: 11 in TN, 8 in KY, 9 in NC, and 25 in NY. Twenty-three (n = 23) agreed to participate in the interview. Eligibility criteria required managers to be currently employed at the designated grocery store as an acting store manager or online ordering department manager or supervisor and over the age of 21. Interviews were conducted at the location of the grocery store of the manager (i.e., on-site at the store), via Zoom, or over the phone. All grocery managers reviewed and provided informed consent prior to participating. Managers who completed interviews received a $100 gift card for participating in the study at the conclusion of the interview. Interviews were conducted until data saturation of themes was reached. This study was approved by the University of Kentucky Institutional Review Board (IRB), study protocol #61763.

A semi-structured interview guide for grocery manager interviews was developed among co-authors with input from the funding agency for this study—Share Our Strength—and is available in the Appendix A. The guide established questions for stores that currently offered online shopping, as well as traditional brick-and-mortar stores not offering online ordering. The online shopping interview guide asked managers about the benefits and challenges of online shopping at their stores. Questions addressed fulfillment processes, product selection, cost, and SNAP redemptions. The brick-and-mortar interview guide asked about the barriers and potential feasibility of incorporating online shopping, the benefits of offering brick-and-mortar-only shopping, and how managers envisioned the future of grocery shopping.

### 2.3. Analyses

All grocery manager interviews were audio recorded and uploaded to the transcription service Rev (Austin, TX, USA). Interviews were transcribed verbatim with redacted identifiable information, such as the store identity, interviewee’s name, and location of residence (i.e., city and/or state of the store) or employment, in order to protect participant identity. There forward, interviews were identified only as a rural or urban grocery store, whether the store currently offered online grocery shopping (delivery or pick-up) and if SNAP benefits were currently accepted in-store and/or with online orders. The produced transcripts allowed investigators to review and conduct thematic analyses. An iterative inductive-deductive approach was employed, and common themes formed the basis of codes which were then analyzed using NVivo software (QSR International Pty Ltd. NVivo, Melbourne, Australia, Release 1.6.1). Using the interview guide for direction, an initial codebook was developed with a priori codes for each shopping type (online and brick-and-mortar) by a qualitative expert (M.B.) in January 2022. One transcript from each retailer type was randomly selected by supplementary coders (R.G. and T.H.) to confirm a priori codes via supportive text segments. Additional emergent codes were identified as a result, and the iterative inductive-deductive process was implored to develop a more accurate codebook. The final codebook was assessed and finalized by other co-authors and investigators. All coders were trained to utilize the codebook. In February 2022, two study team members (R.G. and T.H.) independently coded the interviews using the approved codebooks, and then 30% of the initial coded files were crosschecked for reliability. Two online and one brick-and-mortar initial transcripts were double coded until interrater reliability (IRR) score of ≥0.7 was achieved to ensure consistency before continuation of coding on the remaining transcripts independently. Discrepancies between coding were discussed among coders (R.G. and T.H.) and then were resolved. If resolutions could not be made, a third reviewer (M.B.) then resolved code discrepancies, and themes were agreed upon.

Once coding was complete, coders determined the main themes individually before meeting with the qualitative expert. The two data coders (R.G. and T.H.) assessed coding frequencies among all the transcripts in NVivo and then collapsed the codebook to derive three main themes among online grocery shopping managers and four main themes among brick-and-mortar grocery managers. Subthemes were revealed within each analysis. The data analysis team and the qualitative expert met to agree on the main themes. Final themes and subthemes were then reviewed and agreed upon between all co-authors and investigators.

## 3. Results

The following section includes the results of the semi-structured interviews to include information about the store descriptive in Section 3.1 and each of the overarching themes identified through the data analyses. Tables included in the results section are used to provide quotes to illustrate overarching concepts and examples of thoughts shared within the semi-structured interview.

### 3.1. Rural and Urban Store Descriptives

The final designated stores across each of the participating study states and their geographic area (rural vs. urban counties) are described in Table 1. Investigators made attempts with several grocery managers in each designated location, as capturing diverse perspectives was prioritized to improve external validity measures. While each state included interviews with managers representing both store settings, brick-and-mortar stores (n = 8) consisted of fewer interviews compared to those stores offering online ordering (n = 15).

### 3.2. Managers’ Perceptions of Offering Online Shopping

Three primary themes and five associated subthemes emerged from the interviews with grocery store managers that currently offered online grocery shopping in their stores. Illustrative interview quotes are included in Table 2 to depict emerging themes, along with the number of interviews where each theme was mentioned and the number of times managers addressed the theme during interviews. The main themes included (1) order fulfillment challenges, (2) perceived customer barriers, and (3) perceived customer benefits.

#### 3.2.1. Order Fulfilment Challenges

Order fulfillment challenges generally consisted of factors impacting the grocery store’s operations and their ability to seamlessly provide online grocery services to customers. Due to the relatively early stages of the online grocery medium, coupled with the challenges of the COVID-19 pandemic, this theme was commonly reported among all managers and their respective store locations. Identified challenges included issues related to proper staffing, inadequate technology, and navigating third-party delivery services. One manager noted, “I think the biggest thing that, *[store name]* needs to work on, is the online software, the system that they use. I think it definitely needs to be store specific. It’s aggravating because I want to make it store specific, but that’s obviously not my job. It needs to be because it frustrates the customers. It frustrates us, the people who have to fulfill the orders.” This frustration led to the second theme, perceived customer barriers.

#### 3.2.2. Perceived Customer Barriers

Grocery store managers also shared the perceived customer barriers they have observed since their store began offering online shopping services and to utilizing online grocery shopping services in general. The perceived barriers most frequently reported included store stock, thereby resulting in items the customer may not have preferred, and the fees incurred by customers when utilizing online purchasing platforms. One manager shared, “There is a $4.99 fee at our location, at all *[store name]* locations. That is sometimes a customer concern. Some other retailers may not charge a fee. But we do have a fee currently, and that might be something that changes in the future.”

#### 3.2.3. Perceived Customer Benefits

The most frequently reported online grocery shopping benefit to customers were the convenience and time-saving factors. Reported populations benefitting the most include seniors, homebound individuals, “moms on the go”, mothers with young children, or large organizations placing larger than normal orders. Further, the personal connection online grocery shopping still offers grocery employees and their customers was cited as an additional benefit to engaging in this shopping medium. One manager shared an example of this relationship during their interview, adding, “They do a good job. If you got you good personnel that communicates correctly. And so when they do that job well, and that customer gets used to them, it’s just like that is their personal shopper, and they know their name. They know who they are. They talk to them every week. That means something.”

### 3.3. Brick-and-Mortar Managers’ Barriers and Facilitators to Offering Online Grocery Shopping

Four primary themes and two subthemes emerged from the interviews with grocery store managers at brick-and-mortar locations. The most illustrative quotes are included in Table 3 to depict the emergent themes, as well as how many interviews mentioned each theme and the total number of times each was referenced by managers during interviews. Main themes include (1) online shopping implementation, (2) COVID-19 pandemic impacts, (3) competition with other stores, and (4) benefits of maintaining brick-and-mortar shopping. The two most common themes among brick-and-mortar grocery store managers were implementation of online grocery shopping either within their store, or among grocery stores overall, and the impacts experienced from the COVID-19 pandemic on their store operations. In contrast to online grocery managers, themes related to the pandemic were significantly more prevalent among brick-and-mortar store managers.

#### 3.3.1. Online Shopping Implementation

Managers shared mixed feedback regarding the implementation of online shopping at brick-and-mortar locations. In general, brick-and-mortar managers highlighted the benefits of online shopping, such as the acceptability from a consumer lens for those who are unable to shop in-store. Perceived positive aspects were shared for certain populations, including younger adults, those with families, and those with disabilities. While managers recognized technological advances as being part of the future of the retail grocery environment, they had concerns about the feasibility of adopting online ordering within their stores. Some managers foresaw barriers to even accept online SNAP/EBT in the community their store serves, as a lack of credit card usage by their population was reported. Managers also expressed logistical concerns due to corporate influence, such as a lack of autonomy over the online system and staffing challenges as potential barriers for implementing online ordering services at their stores.

#### 3.3.2. COVID-19 Pandemic Impacts

When managers of brick-and-mortar locations were asked how the COVID-19 pandemic impacted their stores, they primarily described the effects on store operations. Manager and employee roles were adjusted to meet staffing needs and consumer behavior shifted, including a proclivity for panic buying. However, managers reported their efforts to adjust to these changes in real time to ensure the customer experience was a positive one. One manager noted, “I think what we strived to do during the pandemic last year is to be a calming force, if that makes any sense. We had a lot of panic buying, a lot of panic shopping and everything. It just goes right back to the basics of good service.”

#### 3.3.3. Competition with Other Stores

The possibility of competing with other store locations was described by the brick-and-mortar managers as a major concern related to adopting online grocery shopping. Specific store types identified as competitors were beyond traditional grocery store competitors and included discount or dollar stores and supercenters. Managers indicated they would have to fight for their business to meet consumer needs and remain competitive in the retail market. When asked to expand on this, one manager shared, “I would say there is still some loyalty, as far as the change, but I had found that you really got to fight a little harder now. One thing that has changed, pretty much everybody’s in the grocery business now. Twenty-five years ago when I started, *[Supercenter]* was really not a player in the grocery industry per se, but they are now. A lot of your dollar stores…You’ve got a lot more competition as far as that goes, beyond just your traditional competitors.”

#### 3.3.4. Benefits of Maintaining Brick-and-Mortar Shopping

For our sample of brick-and-mortar grocery managers, maintaining current retail capacity was a priority. Managers shared the benefits of in-person shopping and brick-and-mortar operations. They appreciated the personability of in-store shopping and getting to know their customers and families. Many felt that their customers are fairly loyal to their store and in-store shopping and want to maintain that presence in their community. One manager shared an example of this dynamic during their interview, noting, “Well, the being personable, of course. The hand-to-hand connection you have with our customers, because our customers... Mainly our customer quality is because of the friendly atmosphere, the family atmosphere. I mean, that’s what we thrive on. We thrive to be family-like to our customers, and we know some of them intimately. I mean, just, we know their children, we know their children’s children.” Having positive relationships with customers was important to retailers trying to stay competitive with other retailers in their community.

## 4. Discussion

This study’s findings provide a deeper understanding of the experiences of retail grocery managers that serve SNAP shoppers in both rural and urban settings. We add to the current literature by exploring both the barriers and facilitators of online shopping from the retailer’s perspective and the barriers to feasibility of incorporating online services into brick-and-mortar grocery stores in both urban and rural communities. Further, this study aimed to explore the retailer-level attributes the proposed online food retail conceptual framework outlined [13]. This was conducted by examining the experiences and perceptions of individual grocery store managers on how online shopping influences consumer behaviors as they engage in the process from start (selects online retailer) to finish (receives order). However, our findings depict a new perspective within the retailer level of the framework (Figure 2). In particular, marketing and website customization serve as the crux of what retailers provide to customers when they engage in the online shopping phase that comprises the quality of their experience. However, marketing and website customization was indicated by managers as one of the major challenges they experience throughout the online shopping process. Managers specifically mentioned as a challenge listed inventory on the website not in stock in store, as revealed through the order fulfillment challenges theme and technology subtheme. They also recognized promotional or marketing tactics customers experience as a component they had minimal or no knowledge of, revealed by the lack of a major theme in the analysis, instead of a domain they had conscious and explicit control over, which could thereby influence consumer shopping choices. Rather, the perceptions and experiences of grocery managers included in this study reveal the expansive levels within the retail industry, from high-level corporate decision makers to individual store employees fulfilling online orders. These findings reinforce the wide spectrum of points of influence retailers have on impacting consumer behaviors throughout the online shopping process. Figure 2 depicts how and where this study’s findings fit within Khandpur et al.’s framework and propose the distinct differentiation of roles, experiences, and influence within a retailer and corporate structure. These policies, operations, and retailer fulfillment processes cultivate the online grocery shopping landscape and customer experience. Whether engaged at the corporate level or managerial level of control, these factors influence consumer behavior, both directly and indirectly, across the path-to-purchases phases from pre-to post-shopping trip.

In the context of the pandemic, online grocery managers viewed offering customers the ability to order groceries online as convenient, especially if ordering in bulk, and benefiting some vulnerable groups. In contrast, brick-and-mortar managers perceived more negative effects of the pandemic compared to managers of stores offering online grocery services. This is concerning since brick-and-mortar stores are more likely to be located in rural areas, which already experience inequitable food access issues compared to urban areas [23,24]. Managers of brick-and-mortar grocery stores viewed online grocery services as a way to improve the competitiveness of their stores. However, it was stressed throughout interviews with brick-and-mortar store managers that they prided their stores’ ability to keep personal connections with their customers posing the unique juxtaposition many brick-and-mortar store owners and managers are now contemplating as this new grocery shopping medium continues to grow.

Due to the pandemic and supply chain issues, store owners commented on supply stock being a particularly important barrier they faced when offering online services. Perceived customer ordering barriers largely revolved around one main idea—customers were generally unhappy with online substitutions (due to stores’ not having their item in stock) or shopper selections. Managers speculated that this was typically a result of either (1) customer not utilizing the feedback or communication features offered in the grocery shopping platform when they place their online order; or (2) the online platform did not correctly display what was and was not in stock, thus forcing a substitution to be made that they were not happy with. This is an important finding that corporate-level retailers’ roles should consider to improve both the experience customers have with the online platform and the experiences store-level employees have with the customer directly when fulfilling the order.

Another barrier to online grocery shopping included lack of staffing to help fulfill online orders. Specifically, store managers had difficulty retaining staff due to limited wage opportunities, as staff were not as willing to take front-line jobs that could increase their risk of contracting COVID-19 during the pandemic for limited wages. Having to find, train, and then replace staff due to turnover served as a major barrier to seamlessly offering online grocery ordering fulfillment. Additionally, the lack of technological infrastructures that were in place before the pandemic served as a barrier for smaller or rural stores. The amount of funding required to update online ordering platforms, in addition to adhering to the security and compliance requirements for the SNAP OPP, was a barrier to entry for some brick-and-mortar stores. Some store managers felt like the use of SNAP benefits would need to be the main priority when making a switch to online offerings, but the cost of becoming a SNAP-authorized online vendor was a major barrier mentioned whether it was related to the need for more staff, upgrading technology, or increasing their supply in the store initially.

In terms of changes to the retail environment in the community, most store managers noted a change in their customers’ general purchasing behaviors during the pandemic. They stated customers had adopted a more minimalistic approach to interacting with store staff, as well as trying to simplify their shopping experience by minimizing the amount of time shopping in the store. In a broader sense, some customers embraced online shopping due to its convenience and feeling safer rather than shopping in-store. One rural brick-and-mortar grocery store even had some customers call their store, give a store employee their “order”, and pick up their order later on—such as an informal curbside pick-up program. The shopping behavior changes, including a proclivity for panic and bulk buying, among customers seemed to be most apparent in brick-and-mortar stores. However, for stores offering online shopping, there appeared to be more reported issues with the supply chains, staffing, and technology. It is an important finding to note how the pandemic affected grocery stores, grocery managers, and store operations differently based on store type and geographic location.

This study was designed to explore managers’ perception and knowledge of SNAP-customer frequency and interaction with online ordering services, in addition to investigating the experiences and perceptions of online grocery shopping overall among rural and urban store managers. However, this was not a salient theme in our analysis. Among manager interviews offering online ordering, total SNAP-related codes were the least frequently reported code compared to the others. Further, SNAP-related questions in the interview guide did not provide illustrative quotes nor were discussed by store managers in detail. Therefore, it was determined among coders to not include SNAP feedback as a major theme, although it was mentioned during the interviews. A possible explanation is that managers do not know who is a SNAP customer during the shopping process when fulfilling an online grocery order, and therefore, managers had little feedback to provide when asked about the online shopping experience of SNAP customers. This could be a positive factor of the SNAP OPP related to potentially decreasing stigma among SNAP participants, which still needs further investigation, as our study did not assess SNAP customers’ perspectives directly to determine whether they were even aware of this protection in anonymity when shopping online. Furthermore, most grocery managers reported not having access to the purchasing data—either because the online ordering was through a third party or because it was handled at the corporate level. Therefore, although an initial objective of the study was to further understand how SNAP customers may engage with online shopping platforms and any potential barriers they face as the SNAP OPP is rolled out, we learned two valuable insights: (1) progress has been made to protect the identity of customers and potentially reduce the stigma SNAP customers may face when grocery shopping online, and (2) answering this question lies at the corporate level rather than store level, further highlighting the extensive hierarchy of corporate management levels that impact customers’ behaviors when online shopping.

To date, there has been little research investigating retailers’ experience using online grocery services. Integrating grocery store managers’ perceptions of facilitators and barriers to online grocery services with what is already known from customers’ experience purchasing groceries online will help inform public health interventions, retailers’ practices, and local and federal policies for creating a more healthful and equitable online food environment. These efforts will also support expanded food access among low-income consumers and other vulnerable populations with online grocery shopping services.

### 4.1. Implications for Research and Policy

Communication is important for navigating new platforms and purchasing in the virtual marketplace. Previous work exploring online grocery policies and practices of 21 retailers found that none, including those participating in the SNAP OPP, had information about SNAP on the homepage of their website [25]. Additionally, although information about the SNAP OPP and information about retailers was commonly available via state press releases, SNAP agency websites, and state COVID-19 information pages for the 47 states approved for online purchasing, health, and nutrition information was inadequate [26]. Limited access to nutrition information has previously been identified as a challenge on retailer websites [25]. Therefore, enhancing the communication is one simple and low-cost way retailers can make it more known to consumers which retailers accept SNAP benefits online. Coupled with healthy eating messaging, these communication strategies can be increased to support healthier choices among SNAP shoppers in the online marketplace.

Future work can explore how to integrate nutrition information into online purchasing for SNAP participants to enhance dietary outcomes of low-income individuals. Innovative ways to deliver nutrition information online to increase healthy purchases, such as incorporating behavioral nudge messaging strategies and prefilled carts with healthier food and beverage items or alternatives [15,21,27]. Online shopping has the potential to expand food access to low-income consumers living in food deserts in urban and rural areas. Further, the expansion and digitalization of online grocery shopping have the potential to increase the availability and accessibility of food for these populations by providing a wider variety of foods through participating retailers [7,9]. However, before this can happen, this study reveals both logistical and perceived barriers, such as technological and staffing needs, that policymakers should be aware of.

The online ordering process still requires attention and investment from policymakers and corporate retailer leaderships to improve healthy food access for underserved populations, including SNAP participants. A recent draft of the Healthy Meals, Healthy Kids Act now includes policy mandates that would require each state to support at least three authorized retailers to accept Women, Infant, and Children (WIC) EBT online grocery purchases, which would aid in efforts to improve nutritional access among this population [28]. The OPP’s goal is to promote an equitable expansion of healthy food access through online grocery shopping [29], and this study highlights where investments can be made to continue these efforts since addressing the barriers that contribute to food insecurity and limited food access are complex.

### 4.2. Limitations

This study has several limitations, as our study only included 23 interviews across four states, three of which are located in the Southeast region of the U.S. Due to the qualitative design of this study and the nature of qualitative studies, which are designed to explore a specific issue in depth in a specific context, this study lacks the generalizability of other research designs [30]. Additional research on online grocery access in different domestic and international contexts is needed. Future work in the area could benefit from utilizing similar qualitative method approaches to understand how to improve online shopping systems for retailers and consumers. Additionally, views from the store managers who participated in this study may differ from store managers who chose not to—or were unable to—participate in interviews, therefore limiting the generalizability of these results to all multinational grocery store corporations. However, the study is strengthened by its focus on both rural and urban grocery stores, understudied areas, multi-state data collection efforts, double-coding including analyst triangulation to establish credibility of the qualitative data, and reconciling of codes by coders, therefore producing an audit trail to establish confirmability [31].

## 5. Conclusions

This study is one of the first to explore the specific experiences and perceptions of retail grocery store managers when carrying out the online grocery shopping process for customers who utilize this shopping practice. As online grocery shopping continues to expand both by consumers and across the corporate landscape, it is important for input from store-level management to be considered to efficiently implement the technological and operational enhancements that may evolve. The ongoing collection of qualitative data to assess prospective feedback could assist retailers in understanding how to adjust or improve their systems as online shopping continues to grow. Further, although the SNAP OPP is relatively new, this study found promising feasibility and acceptance of this initiative among stores that accept SNAP EBT online. However, more work is warranted to expand awareness of this program among SNAP participants, as online ordering poses to improve food accessibility disparities among underserved and vulnerable populations in the U.S. Future policies should prioritize equitable expansion of online grocery services for all populations as the online grocery landscape evolves.

## Figures and Tables

**Figure 1 nutrients-14-03794-f001:**
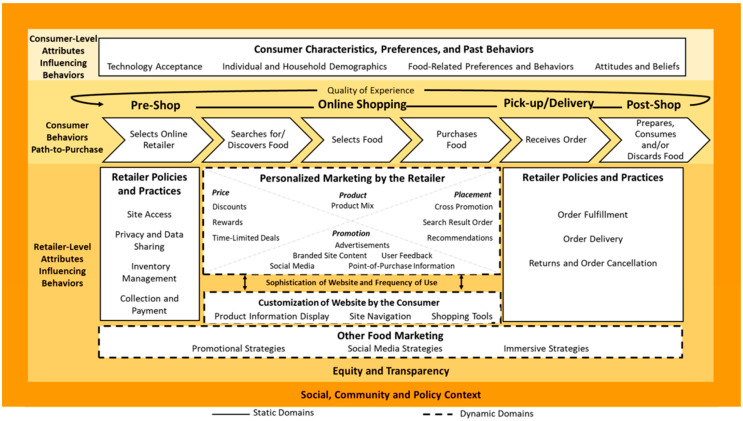
Khandpur et al. proposed online grocery framework. This framework outlines the different levels that comprise online grocery shopping and thereby can influence consumer engagement and utilization of online grocery services.

**Figure 2 nutrients-14-03794-f002:**
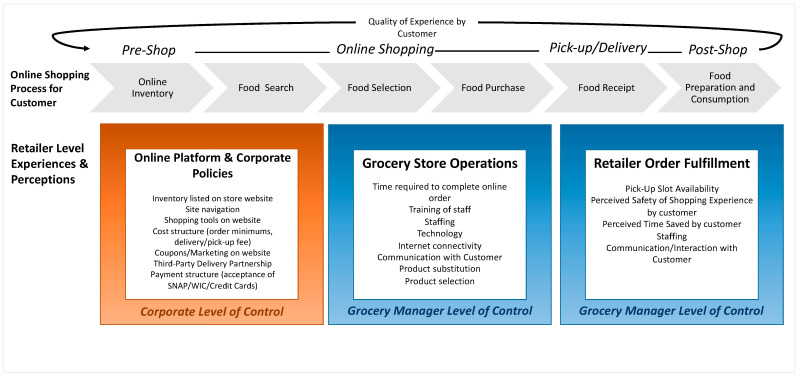
Rural and urban grocery managers’ experiences and perceptions. This study contributes a new lens to consider as the online grocery food environment conceptual framework evolves.

**Table 1 nutrients-14-03794-t001:** Grocery store manager interview store locations and operating capacity by county type.

State	Store Name ^1^	Rural (R)/Urban (U)	Brick and Mortar	Online Ordering
Tennessee (n = 6)				
TN	Store A	R	X	
TN	Store B	U		X
TN	Store C	R	X	
TN	Store D	U		X
TN	Store E	R	X	
TN	Store F	R	X	
Kentucky (n = 5)				
KY	Store A	U		X
KY	Store B	U		X
KY	Store C	U		X
KY	Store D	R		X
KY	Store E	R	X	
North Carolina (n = 6)				
NC	Store A	U		X
NC	Store B	U		X
NC	Store C	R	X	
NC	Store D	R	X	
NC	Store E	R		X
NC	Store F	R		X
New York (n = 6)				
NY	Store A	U	X	
NY	Store B	U		X
NY	Store C	U		X
NY	Store D	U		X
NY	Store E	U		X
NY	Store F	U		X

^1^ Stores A–F designate different store locations within each state and do not correlate to same store locations despite multiplicity of identifier use in table.

**Table 2 nutrients-14-03794-t002:** Primary emergent themes from qualitative analysis among n = 15 grocery store managers offering online grocery shopping to customers.

Theme	Subtheme	Description	Illustrative Quotes (R/U)	# of Interviews That Mention This Theme	# of Times This Code Was Used
Order fulfilment challenges		Store-related challenges to online order fulfillment.	“They purchase so much, like $500, $600 grocery orders, which I think in the store, it would fill up two or maybe three of those regular shopping carts. So if you can just imagine that coming down the check lane.”… “It takes a lot of dedication and focus to make sure that we do our best to make sure every order gets completed in a timely manner, but those things specifically, I would say are out of our control, definitely slow us down.” (U) “We had to get more people in the store trained to pick orders and to do pick up, as more people tried to transition to that.” (R)	17	42
	Staffing	Staffing-related issues contributing to the challenges of online order fulfillment.	“These days, you can’t find good decision-making adults that want this job because they’re only going to make $12 an hour. They can’t feed their family with $12 an hour. So I get a lot of high school kids, a lot of really young college kids and some of them are great. They make great decisions. They’re just really good kids.” (U) “The labor situation is the worst I’ve ever seen it. I’ve worked for *[store name]* for 15 years and there’s nothing even close to what we’re dealing with right now.” (R)	14	37
	Technology	Technology-related issues that may contribute to fulfilling online grocery orders.	“And so right now we’re using tablets to process those payments and *[Store name]* likes to run before they walk sometimes. And the push to get those tablets out was pretty big. And they’re great when they work. But if the wind blows the wrong... they don’t really have great connectivity. They don’t have their own cellular data. So they’re actually pulling off WiFi from the store.” (U) “I think the biggest thing that, I guess, *[store name]* needs to work on, is the online software, the system that they use. I think it definitely needs to be store specific. It’s aggravating because I want to make it store specific, but that’s obviously not my job. It needs to be because it frustrates the customers. It frustrates us, the people who have to fulfill the orders.” (R)	10	25
	Third-party delivery options	Intersection of additional platforms being used for delivery-like services that allow customers to complete an online grocery order that is then delivered by the third-party provider to the customers home (or delivery address of choice).	“There’s not a lot of accountability on *[third-party delivery company]* end when things go wrong”… “Which is very frustrating because if you get on *[Store Name]* website and you go to order delivery, it looks like we’re doing it.” (U) “Well at the volume that *[third-party delivery company]* did for me wasn’t good enough and the cost was too high because I was still paying, between payroll and expenses, about 30 cents on the dollar of everything I buy and then they were charging me 15% of the *(customer)* transaction, so that was bringing down my transaction price. I was working on about 55 cents on the dollar and that was costing me about 45% of investment to do that transaction *(with the third-party delivery company)*. The volume wasn’t there so I gave it up.” (U)	17	54
Perceived customer barriers		Manager perception, or direct feedback from customers, regarding the disadvantages to the online ordering process and use.	“And sometimes customers are okay with not getting an item, or sometimes they just don’t like the item we choose. But that’s one of the biggest obstacles.” (R) “And I guess the app is not tied to our actual on-hand inventory. So it doesn’t know that such and such is out of stock.” (U)	12	30
	Cost	Additional fees incurred due to utilization of the online platform for purchasing groceries.	“There is a 4.99 fee at our location, at *all [store name]* locations. That is sometimes a customer concern. Some other retailers may not charge a fee. But we do have a fee currently, and that might be something that changes in the future.” (R) “The only thing we have in pick up is if your order’s not over $35, it’s a 4.95 charge. That’s the only other charging take up that is there.” (R)	18	47
Perceived customer benefits		Manager perception of the benefits customers receive from their store using an online shopping platform to do their grocery shopping.	“So directly from the customer, we get a lot of your time saver, people with mobility issues that can’t come into the store usually. So those are kind of the easy ones. New moms who have newborns, if they don’t want to bring them into the store, I wish it was around when I had my newborns. So those are kind of the big ones.” (U) “And I also see at this location, what I haven’t seen as much in other locations since then, a lot of times we’ll have organizations that will order their orders online, maybe churches, or daycares, and stuff like that, that’ll order large bulk orders online to pick them up. And like I said, a lot of times it’s just a matter of convenience and a time saver for various different people.” (R)	14	31
	Personal connection	Ability to personally communicate through the online shopping mechanism or platform.	“They do a good job. If you got you good personnel that communicates correctly. And so when they do that job well, and that customer gets used to them, it’s just like that is their personal shopper, and they know their name. They know who they are. They talk to them every week. That means something.” (R) “We do have a good communication feature through our app, though. So they’re able to... I think it’s like a voice-to-text kind of deal and they’re able to be like, ‘Hey, we’re out of this, does this work for you?’ That kind of thing. So that way we do have a really good communication with the people that place the orders.” (U)	17	33

**Table 3 nutrients-14-03794-t003:** Primary emergent themes from qualitative analysis among n = 8 grocery store managers offering brick-and-mortar grocery shopping.

Theme	Subtheme	Description	Illustrative Quotes (R/U)	# of Interviews That Mention This Theme	# of Times This Code Was Used
Online shopping implementation		Feasibility of offering online shopping for the respective store. General feedback from store manager regarding the implementation and act of online grocery shopping.	“In general I think it is a great idea. And I’m looking forward to... Especially if we are able to incorporate EBTs with food stamps into that in a official or legal capacity, I feel like that would be amazing and I feel like that could really boost our business.” (R) “I think it’s the way it’s moving. I don’t think you can’t undo what’s already been done. You can’t go backwards. And so I think that’s where it’s headed. Not only is it convenient for families in general, who are moving in a million different directions, the convenience of quickly ordering your groceries when you’re on lunch break and then quickly grabbing them on your way home from dinner or ball practice or whatever it is. I just think it just makes sense with our current lifestyle right now.” (R)	9	29
	Perceived customer response to offering online ordering	Store manager thoughts on utilization of online ordering if the store was to offer it in the future.	“Personally, I think that with our younger crowd and our almost say our middle-aged crowd, of those that already have say established families, younger families, middle school, high school families. I think they would really enjoy that opportunity.” (R) “So definitely our handicapped customers, I would say, would probably be using it the most, I know that there’s quite a few customers that have wheelchairs, or the elderly. We have currently quite a few people who will sit in our parking lot and call us and give us a list of stuff to pick out for you.”… “So, that probably would be the targeted demographic.” (R)	8	13
	Barriers to implementation and overall change	Logistic or employee barriers that would need to be overcome before attempting to make changes at their store to include offering online shopping.	“Yeah, staffing, because we don’t have the disposable income and payroll to just be able to pay someone to be there extra each day. I know most days we are either sending people home early because they cannot stay or asking people to stay late because we just need some extra people there, unfortunately, and it’s on such a day-by-day basis.” “Yeah, it’s a process. There’s certain things you have to have in place. You have to have your infrastructure, you have to have your tech there. And until you have those things there, you can’t even really.”	9	23
COVID-19 pandemic impacts		Impacts of the COVID-19 pandemic on the store operations (physically or economically), manager roles, or consumer shopping behavior.	“I think what we strived to do during the pandemic last year is to be a calming force, if that makes any sense. We had a lot of panic buying, a lot of panic shopping and everything. It just goes right back to the basics of good service.” “We went days that it was every day, he had to shift and adjust. There was one day that we didn’t have anybody in produce. No one to do produce because we were waiting to hear back about who was good and who wasn’t good. And one day we didn’t have two bag boys.” (R) “And so I just went to *[co-worker]* and I said, I guess we’re baggers this afternoon.” (R)	9	29
Competition with other stores		Assessing the influence of customer changes as a result of other convenience, dollar, and online store offerings or competition.	“I would say there is still some loyalty, as far as the change, but I had found that you really got to fight a little harder now. One thing that has changed, pretty much everybody’s in the grocery business now. Twenty-five years ago when I started, *[Supercenter]* was really not a player in the grocery industry per se, but they are now. A lot of your dollar stores.”… “You’ve got a lot more competition as far as that goes, beyond just your traditional competitors.” (R) “We’re very rural. There’s still a vast amount of people who still do go shopping. So *[Supercenter],* I would say, would be the more... What’s the word? Competitive, I would say, in terms of fighting for our business, the business of our customers. And they have just recently started offering the shopping online.” (R)	7	18
Benefits to maintaining brick-and-mortar shopping		Manager perceived beliefs about the benefits of remaining an in-person-only operation.	“Well, the being personable, of course. The hand-to-hand connection you have with our customers, because our customers... Mainly our customer quality is because of the friendly atmosphere, the family atmosphere. I mean, that’s what we thrive on. We thrive to be family-like to our customers, and we know some of them intimately. I mean, just, we know their children, we know their children’s children.” (R) “Well, you don’t get that one-on-one time either. They come in and talk to you every day. They can tell you their story, their whole life story and all that, and you can’t just walk in. You’ve got to order it online, and then wait until they say it’s ready. You might have something you need to do and your groceries aren’t ready, so what are you going to do?” (R)	5	16

## Data Availability

The data presented in this study are available on request from the corresponding author (R.G.). The data are not publicly available due to protection of participant identities, and all provided data will be de-identified.

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
