# Peer review of "Barriers and Facilitators of Online Grocery Services: Perceptions from Rural and Urban Grocery Store Managers"

_nutrients, 2022, doi:10.3390/nu14183794_

Round 1

Reviewer 1 Report

This is an interesting and thought-provoking article on the barriers and facilitators of online grocery services, particularly focusing on the perceptions from US rural and urban grocery store managers. A qualitative research approach was used (in-depth interviews). This paper deserves publication after some minor adjustments. Mainly, I have two concerns with this current version.

My first concern is that I believe the paper would benefit from having two separate initial sections. The first (the introduction) is already there, but should be brief and focus on the research gap and research question only. A second one (with the literature review) should be included (with some eventual sub-sections/topics). More citations are welcome here.

A second concern deals with the generalizability of the results. Are the results from the US equivalent to other cultures? Some benchmarking with other countries is needed. Also, more references should be included, particularly from the COVID-19 online grocery context, for example:

Gruntkowski, L., & Martinez, L.F. (2022). Online grocery shopping in Germany: Assessing the impact of COVID-19. Journal of Theoretical and Applied Electronic Commerce Research, 17(3), 984-1002. https://doi.org/10.3390/jtaer17030050

The method, results, and (mostly) discussion sections are very solid, I have no further recommendations. Very nice work!

Good luck with the revision!

Reviewer 2 Report

The article is very well written and structured and intends to explore the possibilities of expansion of online grocery also as a result of the pandemic. in particular it looks at the supply side with interviews with managers within a specific programme (Snap).

The first recommendation I make is to explain for non-US readers better some practices starting with the programme itself in addition to the quote (number two) a note or parenthesis explaining what it consists of would be helpful. see line 44.

I believe that apart from the availability of additional material an article should stand on its own, therefore it should be clear and self-sufficient without necessarily requiring consultation of the material.

In the same vein, I recommend adding the initials of the states after the first citation, so about line 120 -122.

The introductory part is very focused but in my opinion in order to avoid the article becoming a case study it would be necessary to broaden the view and in particular to enrich the literature part. therefore a somewhat deeper analysis of the existing literature is needed, e.g.

Martín, J. C., Pagliara, F., & Román, C. (2019). The research topics on e-grocery: Trends and existing gaps. Sustainability, 11(2), 321.

As a perishable, perishable experience good with a not particularly high unit cost, in which therefore the delivery component plays a decisive role for both demand and supply, the literature on delivery modes related to bar/online purchasing should also be read and cited

the 2021 special issue E-groceries, digitisation and sustainability in Research in Transportation Economics, 87

In general, for example in Europe, the work of Marcucci & Gatta, that of Lagorio & Pinto, or, on the planning aspect, of Bjorgen...

I recommend introducing sections 2.1 and 2.2 etc. on line 112

I would replace Table One - to be put in the appendix possibly with this degree of detail - with a more effective visualisation of the data. In any case it should be commented on.

The same can be said of Tables Two and Three, where the numbers in the last two columns would be better illustrated with a graph.Here too the commentary should be expanded and the usefulness of the quotations in full illustrated. In this respect, objectively speaking, the length of the quotations is a bit redundant in my opinion. Unless you want to introduce a content analysis, I think it would be more useful to summarise the concept instead of quoting directly in inverted commas.

Check the figure captions: imprecise the first one lacks the year and in any case both are too long. I would expand the comment in the text rather than condense it cryptically in the caption

Check title 3.2 and line 200

Check fullfilment word consistency (one or two l?) lines 206 207

Round 2

Reviewer 2 Report

Dear authors, first of all I would like to thank you for your reactions to our comments.

Regarding your responses, and the new version of the manuscript, I would like to point out:

1. That there is rightly a lack of specific literature on the subject - hence the originality of your article - but this does not, in my opinion, preclude you from expanding (at least in the introduction, as suggested by the first reviewer) the view on the subject with citations of works that have dealt with the problem in different ways in different countries in different periods, etc. Moreover, for example, there is a lot on the demand side while the literature is lacking on the supply side; it would be interesting to specify this precisely to make it clear that you, who deal with the supply side, are innovative. 

In this regard I think I have already pointed out a few contributions 

Martín, J. C., Pagliara, F., & Román, C. (2019). The research topics on e-grocery: Trends and existing gaps. Sustainability, 11(2), 321.

Special Issue:

E-groceries, digitisation and sustainability in Research in Transportation Economics, 87.

In general, for example in Europe, the work of Marcucci & Gatta, that of Lagorio & Pinto, or, on the planning aspect, of Bjorgen...

Covid and e-grocery are analysed by Sernicola, F., Maltese, I., Gatta, V., Iannaccone, G., & Marcucci, E. (2020). Impact of lockdown on Italians' spending: what future for e-grocery?

and many others in these last 2 years

2. I suggested putting two lines between the title and the subtitle to explain what follows in the section

EXAMPLE:

"2. Materials and Methods 

In this section ....2.1 says that ...while 2.2 illustrates...

2.1. Study Sites" 

3. It is fine to keep the tables so detailed, but in my opinion a comment is needed, i.e. writing that a theme in the interviews occurs 17 times and another only 3 times without a comment and without referring to what is written in the table makes these tables really only aesthetic. 

4. Similarly, I consider the sentence in the caption to be insufficient - I would prefer to see it expanded in the text in conjunction with a reduction that immediately clarifies what the usefulness and message of the figure is (title, as for the tables). 

5. In my opinion, qualitative studies can be replicable, in terms of approach and policy implications: could it be helpful for you in order to broaden a bit the discussion and the conclusions?

6. I would move the text of 4.2 and 3 in the conclusions. 
